# Association of Insulin-like Growth Factor-1 with Bone Mineral Density in Survivors of Childhood Acute Leukemia

**DOI:** 10.3390/cancers16071296

**Published:** 2024-03-27

**Authors:** Seulki Kim, Jae Won Yoo, Jae Wook Lee, Min Ho Jung, Bin Cho, Byng-Kyu Suh, Moon Bae Ahn, Nack-Gyun Chung

**Affiliations:** Department of Pediatrics, College of Medicine, The Catholic University of Korea, Seoul 06591, Republic of Korea; seulki12633@gmail.com (S.K.); hoiring0209@gmail.com (J.W.Y.); dashwood@catholic.ac.kr (J.W.L.); jmhpe@catholic.ac.kr (M.H.J.); chobinkr@catholic.ac.kr (B.C.); suhbk@catholic.ac.kr (B.-K.S.)

**Keywords:** acute leukemia, bone mineral density, insulin-like growth factor-1, vertebral fracture

## Abstract

**Simple Summary:**

This study investigated bone mineral deficits in children who survived childhood acute leukemia and analyzed the association of the insulin-like growth factor-1 (IGF-1) level with bone mineral density (BMD) status. Among 214 enrolled patients, 15% showed a low BMD with significant differences in IGF-1 levels. Higher IGF-1 levels were associated with a lower risk of low BMD, suggesting its potential as a valuable tool for assessing osteoporosis in survivors of childhood acute leukemia.

**Abstract:**

In this study, we investigated bone mineral deficits in children who survived childhood acute leukemia and explored the association between the insulin-like growth factor-1 (IGF-1) level and bone mineral density (BMD). This retrospective analysis enrolled 214 patients treated for acute leukemia, measuring various factors including height, weight, body mass index (BMI), and lumbar spine BMD after the end of treatment. The study found an overall prevalence of low BMD in 15% of participants. Notably, IGF-1 levels were significantly different between patients with low BMD and those with normal BMD, and correlation analyses revealed associations of the IGF-1 level and BMI with lumbar spine BMD. Regression analyses further supported this relationship, suggesting that higher IGF-1 levels were associated with a decreased risk of low BMD. The study findings suggest that IGF-1 may serve as a valuable tool for evaluating and predicting osteoporosis in survivors of childhood acute leukemia.

## 1. Introduction

Acute leukemia is the most common malignancy in childhood [1]. With evolving therapeutic regimens and supportive care, there has been an improvement in the cure rate; however, quality of life after treatment has become a major issue. Impaired bone health is an important problem for survivors of childhood acute leukemia. Bone fragility often originates from multifactorial factors, including the disease itself, treatment, or lifestyle changes during and after leukemia treatment. Problems related to bone metabolism that occur during leukemia treatment and disease-associated factors themselves can cause decreased bone mass and the degradation of bone tissue, leading to heightened bone fragility and susceptibility to fracture and causing growth retardation, resulting in a short stature, which can be a lifelong problem [2,3].

Bone fragility in childhood cancer survivors might occur because of increased bone resorption triggered by cytokines released by leukemic cells; medical therapy, especially glucocorticoids; and comorbidities, such as malnutrition and consequent immobilization [2,3,4]. According to previous studies, children have a much greater fracture risk during and shortly after leukemia treatment than non-diseased children [5,6,7]. Therefore, it is important to evaluate bone health status and predict the risk of fracture in survivors of childhood acute leukemia.

To evaluate bone health, a physical examination should encompass height and weight for body mass index, and it should also involve determining any changes in height velocity in children and adolescents. Additionally, it is important to assess levels of calcium, phosphorus, alkaline phosphatase, vitamin D, and parathyroid hormone. Measuring bone turnover markers such as N-terminal telopeptide of type 1 collagen, pyridinoline, or osteocalcin also helps determine changes in bone metabolism. Measuring bone mineral density (BMD) is the most widely used technique for monitoring bone health status [8,9].

BMD, measured using dual-energy X-ray absorptiometry (DXA) scans, is commonly used to evaluate skeletal health and define osteoporosis, which is indicated by fractures that arise with clinical significance; however, BMD alone may underestimate fracture risks [10,11]. Several factors can potentially disrupt measurements, including movement during the measurement process, scoliosis, and the presence of metalwork, all of which may complicate interpretation [8]. Therefore, surrogate markers suitable for supplementing BMD insensitivity and assessing fracture risk have been studied.

In several studies on adults, insulin-like growth factor-1 (IGF-1), which is known to affect bone metabolism by activating bone remodeling and exerting anabolic effects on bone tissue, has been shown to be a useful tool for assessing fracture risks [12,13,14]. The growth hormone (GH)/IGF-1 axis is one of the most important axes contributing to skeletal health. GH secreted by the pituitary gland regulates the production of IGF-1 from hepatic tissue and its serum concentration and regulates bone metabolism through its direct effects on osteoblasts and osteoclasts. Bone growth and BMD are considerably affected by GH and IGF-1 [15]. Additionally, a few studies have shown a significant positive association between IGF-1 levels and bone mineral accrual in healthy children [16,17,18]. However, research on the relationship between serum IGF-1 levels and bone status in survivors of childhood acute leukemia is lacking.

Therefore, this study investigated the relationship between serum IGF-1 levels and BMD and evaluated the effectiveness of IGF-1 as an indicator of bone health in survivors of childhood acute leukemia. 

## 2. Materials and Methods

### 2.1. Patients

Patients diagnosed with acute leukemia at the Department of Pediatrics, Seoul St. Mary’s Hospital of Catholic University, between February 2000 and February 2020 were screened. Acute leukemia comprises acute lymphoblastic leukemia, acute myeloid leukemia, and juvenile myelomonocytic leukemia. Among them, 279 visited the pediatric endocrine department after leukemia treatment. The therapeutic approach to acute leukemia is contingent upon the specific subtype and risk assessment. Typical modalities encompass immunotherapy, chemotherapy, and radiation therapy. We included patients aged 2–18 years; however, we excluded patients with endocrine diseases, such as hypothyroidism, GH deficiency, and hypopituitarism, or diseases with chromosomal abnormalities, such as Down syndrome, Klinefelter syndrome, and Turner syndrome. Patients experiencing a leukemia relapse during the study period were excluded from the study. Patients without BMD or serum IGF-1 levels were also excluded. In total, 120 male and 94 female participants were included in this study.

### 2.2. Clinical Data

The baseline characteristics of the patients, including age, sex, type of acute leukemia, and treatment, were collected from their medical records. Anthropometric measurements, such as height, weight, body mass index (BMI) (kg/m^2^), and pubertal status, were obtained. Height was measured using a Harpenden Stadiometer (Holtain Ltd., Wales, UK), and weight was measured using a digital scale (Simple Weighing Scale, Cas^®^, Seoul, Republic of Korea). Pubertal staging was assessed by two pediatric endocrinologists according to the Tanner stage criteria based on breast development in girls and testicular volume in boys. All anthropometric parameters were calculated according to age- and sex-matched standard deviation scores (SDSs) using the 2017 Korean National Growth Charts for children and adolescents [19].

### 2.3. Biochemical Parameters

Blood samples were obtained on the same day as the anthropometric parameters and radiological examinations by vein puncture when the children visited the pediatric endocrine department. All serum samples were stored at −70 °C until analysis. Serum IGF-1 and IGF-binding protein-3 (IGFBP-3) levels were measured using a radioimmunoassay kit (Automatic Gamma-10 counter, Shinjin Medix, Goyang, Republic of Korea). Serum IGF-1 and IGFBP-3 levels were converted to SDS values according to the age and sex of Korean children and adolescents [20].

Calcium (absorbance assay, Roche, Basel, Switzerland), phosphorus (molybdate UV, Rapikit, Chandigarh, India), alkaline phosphatase (colorimetric assay according to the International Federation of Clinical Chemists, Roche, Basel, Switzerland), bone-specific alkaline phosphatase (colorimetric assay according to the International Federation of Clinical Chemists, Roche), total 25-hydroxyvitamin D (chemiluminescence immunoassay, DiaSorin, Saluggia, Italy), and intact parathyroid hormone (immunoradiometric assay, DiaSorin) levels were assessed in serum samples with pediatric reference ranges [21,22].

### 2.4. Radiological Data

Dorsal and lumbar spine radiography was used to detect deformities in the vertebral bodies (T5–L5) and assess vertebral fractures (VFs). A VF diagnosis was established when a reduction in vertebral height exceeded 20%, which was determined by two pediatric endocrinologists using a modified Genant semiquantitative technique [23].

Lumbar spine BMD (LSBMD) was measured in the anterior–posterior direction (L1–L4) using DXA (Horizon W DXA system^®^, Hologic Corp., Marlborough, MA, USA) carried out by a single radiographer in charge of BMD measurements. A two-dimensional computation of bone mineral content (g/cm^2^) was performed and converted to an SDS according to age- and sex-matched Korean subjects in a standardized normal distribution [24]. Low BMD was defined as an LSBMD SDS of −2.0 or less according to reference values by age, sex, or pubertal status.

### 2.5. Statistical Analysis

Statistical analyses were performed using IBM SPSS software (version 22.0; IBM Corp^®^, Armonk, NY, USA). Data are expressed as means ± standard deviations for normally distributed values or as numbers (percentages). For the comparison of two groups, an independent *t*-test or χ^2^ test was performed when a normal distribution was assumed, and the Mann–Whitney test or Fisher’s exact test was used when a normal distribution could not be assumed. Spearman’s correlation coefficient was used to describe the association between single variables. Univariate and multivariate logistic regression analyses were used to evaluate the potential risk factors of low BMD incidence, with odds ratios (ORs) and 95% confidence intervals (CIs). Univariate and multivariate linear regression analyses were used to determine the beta coefficient (β) and 95% CI of factors associated with BMD. For all analyses, statistical significance was set at *p* < 0.05.

## 3. Results

### 3.1. Baseline Characteristics of Patients

Table 1 shows a comparison of the characteristics of the male and female study participants. In total, 214 children (120 boys and 94 girls) were included in this study. No significant differences were observed between male and female participants. The mean age at diagnosis with acute leukemia for boys and girls was 8.31 ± 4.093 and 8.47 ± 4.415 years, respectively. The majority of patients had lymphoblastic leukemia, including 92 (76.7%) boys and 69 (73.4%) girls. Fifty-eight (55.2%) boys and forty-seven (50%) girls underwent bone marrow transplantation. The mean age at the DXA examination was 12.50 ± 3.335 years for boys and 12.65 ± 3.403 years for girls. Fifteen (12.5%) boys and seventeen (18.15%) girls had low BMD; thirty (25.2%) boys and twenty-five (26.9%) girls had a VF.

### 3.2. Association of LSBMD SDS with Baseline Characteristics, Bone Markers, and IGF-1

A correlation analysis revealed a relationship between the LSBMD SDS and other factors. The LSBMD SDS was significantly correlated with the BMI SDS at the time of leukemia diagnosis and the time of the DXA examination (rho = 0.230, *p* = 0.001 and rho = 0.356, *p* < 0.001, respectively). Also, IGF-1 SDS and IGFBP3 SDS were positively correlated with the LSBMD SDS (rho = 0.242, *p* < 0.001 and rho = 0.145, *p* = 0.038, respectively) (Figure 1). However, the LSBMD SDS was negatively correlated with age at the time of leukemia diagnosis (rho = −0.234, *p* = 0.001). Calcium and 25-hydroxyvitamin D levels did not show a significant relationship with the LSBMD SDS in this study. Linear regression analyses were performed to determine the factors associated with the LSBMD SDS (Table 2). A multivariate linear regression showed that age at the time of leukemia diagnosis was negatively correlated with the LSBMD SDS (β = −0.049, *p* = 0.043); BMI SDS and IGF-1 SDS at the time of the DXA examination were positively associated with the LSBMD SDS (β = 0.244, *p* < 0.001 and β = 0.124, *p* = 0.041, respectively) (Table 2).

### 3.3. Risk Factors Associated with Low BMD

The age at the time of leukemia diagnosis and treatment, anthropometric factors, and biochemical factors of children with low BMD were compared with those of children without low BMD (Table 3). The BMI SDS was significantly lower in children with low BMD than in those without low BMD (−0.838 ± 1.936 vs. 0.168 ± 2.088, *p* = 0.012). Furthermore, the IGF-1 SDS was significantly lower in children with low BMD than in those without low BMD (−1.23 ± 0.817 vs. −0.72 ± 0.779, *p* = 0.001). However, the age at the time of leukemia diagnosis, transplantation, total body irradiation (TBI), and calcium and 25-hydroxyvitamin D levels showed no significant differences between the two groups. Binomial logistic regression analyses were performed to determine the risk factors associated with low BMD (Table 4). A univariate analysis was conducted to assess age at the time of leukemia diagnosis; the BMD SDS at the time of leukemia diagnosis; transplantation; TBI; the BMI SDS at the time of the DXA examination; and calcium, 25-hydroxyvitamin D, IGF-1 SDS, and IGFBP-3 SDS levels at the time of the DXA examination. Transplantation (OR [95% CI], 2.227 [1.1015–4.886], *p* = 0.046), the BMI SDS at the time of DXA (0.64 [0.486–0.844], *p* = 0.002), and the IGF-1 SDS at the time of DXA (0.759 [0.599–0.963], *p* = 0.023) were associated with low BMD. After adjusting for the abovementioned factors, the BMI SDS at the time of the DXA examination (0.674 [0.504–0.901], *p* = 0.008) and the IGF-1 SDS at the time of the DXA examination (0.762 [0.584–0.994], *p* = 0.045) remained significantly associated with low BMD in this final multivariate analysis model.

## 4. Discussion

Patients treated for acute leukemia may develop various complications after therapy completion owning to the treatment modalities used, such as chemo- or radiotherapy, bone marrow transplantation, and other factors, including nutritional imbalance and changes in lifestyle with reduced exercise [3,25] (Appendix A). After acute leukemia treatment, most patients could experience endocrine and metabolic complications impacting their quality of life throughout their lifetime [26].

Acute leukemia treatment combined with corticosteroid and chemo-immunotherapy can cause metabolic alterations to develop. Hyperglycemia is a common metabolic alteration, increasing insulin resistance and leading to diabetes. Abdominal obesity, dyslipidemia, and high blood pressure, which are risk factors for metabolic syndrome, are commonly occurring complications during and after leukemia treatment. Hypothalamic–pituitary dysfunction, adrenal insufficiency, and thyroid disorders commonly occur as well. Osteoporosis due to changes in bone metabolism is also a complication that can arise in survivors of childhood acute leukemia [26].

To the best of our current knowledge, research investigating the correlation between fracture recurrence and the relapse of acute leukemia is lacking. However, impaired bone health is a critical problem in survivors of childhood acute leukemia [2,3]. Bone fragility in childhood can lead to pathological conditions, such as various types of fractures, including VF, osteoporosis, or growth failure, which can lead to problems throughout an individual’s lifetime [2,3,25]. According to our findings, the incidences of low BMD and VF were 15% and 25.7%, respectively, in the study population. These results are lower than those reported in previous studies [7,27] because previous studies focused on children undergoing treatment from the time of acute leukemia diagnosis, whereas our study targeted children who had completed their treatment. The use of a significant amount of glucocorticoids in the early stages of treatment affects bone metabolism and is associated with a higher risk of fractures [6,27].

Several methods have been devised to identify risk factors for bone diseases and fractures. As DXA provides reliable and precise data, it is widely used in children or adolescents [8]. Moreover, DXA may be indicated when a child is considered for an intervention to decrease fracture risk [10,11]. However, DXA derives BMD by calculating only bone mass and the projected area of bone without considering bone depth, and caution should be exercised while interpreting these results with regard to children with short stature or delayed puberty [8,10,11]. The radiation exposure associated with a DXA scan is minimal, less than that from a standard chest radiograph; however, concerns arise from potential cumulative radiation exposure, which is not negligible as children or adolescents frequently undergo various imaging studies during the course of their disease and treatment. Reduced BMD has well-established clinical significance as it is associated with the risk of fractures; however, it cannot be used to diagnose osteoporosis [10,11]. According to the International Society for Clinical Densitometry, pediatric osteoporosis is characterized by either (1) the presence of one or more VFs without local disease or trauma or (2) a clinically significant fracture history involving two or more long bone fractures by age 10 or three or more long bone fractures up to age 19 in combination with a BMD SDS of ≤−2.0. Low BMD is defined as a BMD SDS of −2.0 or lower, based on reference values considering sex, age, or pubertal status. It is important to note that fractures may still occur even in cases in which the BMD result does not indicate a low BMD [10]. Children or adolescents with normal BMD values may be overlooked because most children or adolescents with VF do not report any signs of bone pain. Hence, spine radiography to evaluate VF and DXA scans should be performed concurrently to compensate for each other for a precise interpretation of pediatric osteoporosis [10,11].

IGF-1 is an important regulator of skeletal homeostasis, including bone modeling and remodeling, by stimulating the differentiation of osteoblasts and osteoclasts. IGF-1 contributes to linear and radial bone growth as well as the maintenance of bone mass [15]. Although the association between serum IGF-1 levels and BMD is debatable, our study results show that serum IGF-1 levels may serve as an indicator of bone health status in survivors of childhood acute leukemia. Serum IGF-1 levels were significantly correlated with LSBMD, and low serum IGF-1 levels were associated with an increased risk of low BMD in our study population. Our findings are consistent with the results of previous studies that found relationships between low IGF-1 levels, osteoporosis, and osteopenia [12,28]. IGF-1 levels are useful tools for evaluating the risk of fractures in adults and children with osteoporosis [14,16,28,29,30]. IGF-1 is produced in virtually all tissues in response to pituitary-secreted GH. Most serum IGF-1 is produced in the liver, and bone is a major depository organ for IGF-1 [15]. IGF-1 levels are influenced by several factors, including sex, age, nutritional status, disease status, and pubertal status [15]. Therefore, for a reasonable interpretation of IGF-1 levels, researchers should consider the patient’s condition. Hence, we calculated the Z-scores for IGF-1 from average IGF-1 levels in normal age- and sex-matched Korean subjects using a standardized normal distribution.

Targeting the GH/IGF-1 axis to ensure optimal skeletal health in children, especially in those who survive acute leukemia, is important. GH treatment (GHT) could help increase longitudinal growth and bone acquisition in short-stature children with GH deficiency (GHD) or normal GH secretion [31,32]. Several studies have evaluated the efficacy and safety of GHT, and most have shown that GHT is safe and effective [33]. In childhood cancer survivors, GHT either leads to height improvement or shows no difference in the risk of metabolic conditions, including type 2 diabetes mellitus, dyslipidemia, or metabolic syndrome [34,35]. Moreover, there is no significant association between GHT and the risk of developing a secondary tumor or tumor recurrence in childhood cancer survivors [34,35]. Based on its safety and efficacy, GHT was attempted in patients with chronic disease, and it was found that GHT effectively influences bone metabolism. In studies of children with chronic kidney disease, GHT led to an increase in BMD in parallel with an increase in muscle mass, primarily by increasing serum IGF-1 levels [36,37]. Furthermore, according to an Endocrine Society Clinical Practice Guideline, GHT is recommended for childhood cancer survivors with GHD when the patient has been disease-free for 1 year or has a chronic stable disease condition even though the patient may not be disease-free for their final adult height [38].

Preventing fractures requires the early prediction of risk and prompt treatment. In cases in which IGF-1 levels are low or decreasing, supplementing with adequate doses of calcium and vitamin D, combined with regular exercise, could be beneficial in the prevention of future fractures. Also, osteoporotic medications can be considered when pediatric treatment for osteoporosis is required [39]. Childhood cancer survivors with osteoporosis may be indicated for medical intervention when improvement is not anticipated despite attempting conservative treatments aimed at optimizing bone health, including sufficient nutrition, adequate body weight, and physical activity, or when they have severe bone pain [11]. Bisphosphonates are considered a treatment option for pediatric osteoporosis and have been widely used in children with osteogenesis imperfecta, cerebral palsy, and connective tissue disease, acting by inhibiting osteoclast-mediated bone resorption and preventing osteoblast or osteocyte apoptosis [40]. Although bisphosphonates have been reported to increase BMD and decrease pain or fractures in children, their use in children remains controversial owing to insufficient evidence of their long-term efficacy and safety, along with concerns about potential adverse effects [41]. Several studies on the use of GHT for osteoporosis in adults have shown that GH may reduce fracture risk, although GHT does not necessarily improve bone density [42,43]. GH and IGF-1 each play a role in stimulating both bone formation and resorption, exerting a significant impact on skeletal metabolism. In addition, GH and IGF-1 exhibit extraskeletal effects, including enhancing muscle mass and strength, implying their crucial roles in decreasing fracture risk [44,45]. Therefore, GHT could be considered a potential additive treatment for osteoporosis in childhood cancer survivors with low IGF-1 levels. However, large-scale prospective, controlled, long-term studies are needed to determine the effect of GH on bone mass and fracture risk in childhood cancer survivors.

This study had some limitations which are mainly related to its retrospective nature. First, this study was conducted at a single center. Only children or adolescents who visited the pediatric endocrine department after leukemia treatment were included, leading to a limited number of enrolled patients. Hence, sufficient clinical and biochemical data may not have been available at the time of cancer diagnosis. A comparison of data at leukemia diagnosis and treatment completion might yield more reliable results. Subsequent studies should focus on comparing the patterns of BMD and IGF-1 changes. However, the accuracy of this study can be considered a strength as height and weight were recorded based on direct measurements taken during hospital visits rather than relying on self-reported data. Second, although we calculated the IGF-1 SDS, this study did not consider pubertal status, sex hormones, and nutritional status, which might affect IGF-1 levels and BMD. Therefore, these factors that may affect IGF-1 levels should be considered in future studies.

## 5. Conclusions

Many childhood acute leukemia survivors experience bone mineral deficits. IGF-1 status may be a prognostic factor associated with bone acquisition and the future occurrence of fractures. IGF-1 status should be monitored with caution in the follow-up of childhood cancer survivors. Moreover, GHT might be considered an additive treatment for osteoporosis in childhood cancer survivors with low IGF-1 levels. Further multicenter studies involving large-scale cohorts are needed to confirm the role of IGF-1 in preventing bone mineral deficits.

## Figures and Tables

**Figure 1 cancers-16-01296-f001:**
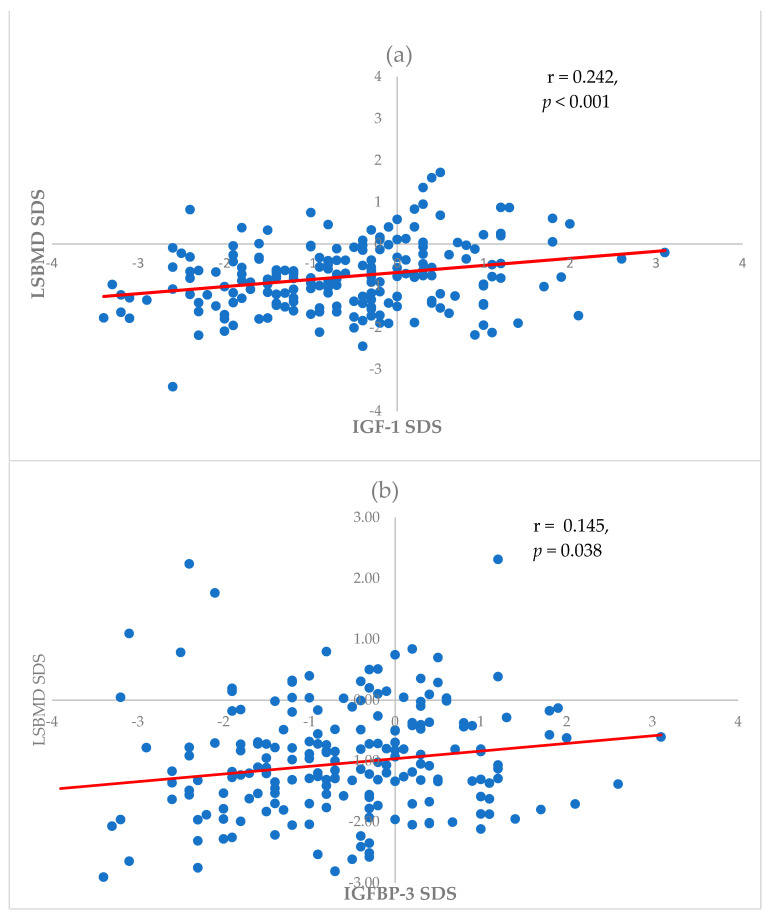
Association of LSBMD SDS with (**a**) IGF-1 SDS and (**b**) IGFBP-3 SDS. IGF, insulin-like growth factor; IGFBP, insulin-like growth factor-binding protein.

**Table 1 cancers-16-01296-t001:** Baseline characteristics of patients.

	Male (*n* = 120)	Female (*n* = 94)	*p*-Value
At leukemia diagnosis			
Age, year	8.31 ± 4.093	8.47 ± 4.415	0.777
Height SDS	0.31 ± 2.518	0.05 ± 2.382	0.461
Weight SDS	0.06 ± 2.392	0.10 ± 1.952	0.902
BMI SDS	−0.05 ± 1.371	−0.05 ± 1.437	0.980
Leukemia category			
Lymphoblastic	92 (76.7%)	69 (73.4%)	0.562
Myeloid	25 (20.8%)	25 (26.6%)	0.323
Juvenile myelomonocytic	3 (2.5%)	0 (0)	0.123
Transplantation, n (%)	58 (55.2%)	47 (50%)	0.891
Total body irradiation	43 (36.4%)	32 (35.2%)	0.849
At DXA examination			
Age	12.50 ± 3.335	12.65 ± 3.403	0.763
BMI SDS	0.018 ± 2.450	0.018 ± 1.540	0.998
IGF-1 SDS	−0.651 ± 0.769	−0.971 ± 0.813	0.040
IGFBP3 SDS	−0.897 ± 1.215	−1.142 ± 1.110	0.140
aBMD, g/cm^2^	0.825 ± 0.189	0.853 ± 0.213	0.407
LSBMD SDS	−0.683 ± 1.556	−0.616 ± 1.363	0.973
Low BMD, n (%)	15 (12.5%)	17 (18.1%)	0.256
VF, n (%)	30 (25.2%)	25 (26.9%)	0.783

BMI, body mass index; SDS, standard deviation score; DXA, dual-energy X-ray absorptiometry; IGF-1, insulin-like growth factor 1; IGFBP3, insulin-like growth factor 3; aBMD, areal bone mineral density; LSBMD, lumbar spine bone mineral density; VF, vertebral fracture.

**Table 2 cancers-16-01296-t002:** Univariate and multivariate regression analyses of factors associated with lumbar spine bone mineral density standard deviation score.

Risk Factors	Univariate			Multivariate		
	β	SE	*p*-Value	β	SE	*p*-Value
At leukemia diagnosis						
Age	−0.06	0.024	0.012	−0.049	0.024	0.043
BMI SDS	0.125	0.05	0.014			
Transplantation	−0.497	0.199	0.013			
TBI	−0.529	0.207	0.011			
At DXA examination						
BMI SDS	0.315	0.062	<0.001	0.244	0.065	<0.001
Calcium	0.415	0.27	0.125			
25OHD	0.003	0.011	0.967			
IGF-1 SDS	0.154	0.059	0.010	0.124	0.06	0.041
IGFBP-3 SDS	0.054	0.057	0.347			

β, beta coefficient; 25OHD, 25-hydroxyvitamin D; SE, standard error; TBI, total body irradiation.

**Table 3 cancers-16-01296-t003:** Clinical data of children with and without low bone mineral density.

	LSBMD SDS ≤ −2.0 (*n* = 32)	LSBMD SDS > −2.0 (*n* = 182)	*p*-Value
Age at leukemia diagnosis	9.53 ± 3.750	8.18 ± 4.247	0.092
Transplantation	21 (65.6%)	84 (46.2%)	0.055
TBI	16 (51.6%)	59 (33.1%)	0.067
Height SDS	−1.278 ± 1.450	−0.235 ± 1.340	<0.001
Weight SDS	−1.319 ± 1.742	0.028 ± 1.860	<0.001
BMI SDS	−0.838 ± 1.936	0.168 ± 2.088	0.012
Calcium (mg/dL)	9.30 ± 0.420	9.43 ± 0.376	0.075
Phosphorus (mg/dL)	4.22 ± 0.687	4.47 ± 0.646	0.043
ALP (U/L)	158.28 ± 88.935	167.16 ± 94.497	0.639
Bone-specific ALP (µg/L)	60.83 ± 43.205	60.12 ± 39.029	0.960
Calcitriol (ng/mL)	109.62 ± 452.432	21.04 ± 10.053	0.337
PTH (pg/mL)	39.93 ± 17.913	45.87 ± 25.528	0.461
IGF-1 SDS	−1.23 ± 0.817	−0.72 ± 0.779	0.001
IGFBP-3 SDS	−1.16 ± 1.732	−0.92 ± 1.777	0.494
aBMD (g/cm^2^)	0.702 ± 0.151	0.860 ± 0.199	<0.001

ALP, alkaline phosphatase; PTH, parathyroid hormone.

**Table 4 cancers-16-01296-t004:** Univariate and multivariate regression analyses of factors associated with low bone mineral density.

Risk Factors	Univariate			Multivariate		
	Exp(B)	95% CI	*p*-Value	Exp(B)	95% CI	*p*-Value
At leukemia diagnosis						
Age	1.081	0.987–1.183	0.092			
BMI SDS	0.098	0.800–1.200	0.842			
Transplantation	2.227	1.015–4.886	0.046	1.493	0.595–3.482	0.419
TBI	2.151	0.215–1.004	0.051			
At DXA examination						
BMI SDS	0.64	0.486–0.844	0.002	0.674	0.504–0.901	0.008
Calcium	0.41	0.153–1.101	0.077			
25OHD	0.98	0.935–1.027	0.403			
IGF-1 SDS	0.759	0.599–0.963	0.023	0.762	0.584–0.994	0.045
IGFBP-3 SDS	0.921	0.730–1.163	0.490			

CI, confidence interval.

## Data Availability

The data presented in this study are available upon request from the corresponding author. The data are not publicly available due to the privacy of the research participants.

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
