# Peer review of "Association of Insulin-like Growth Factor-1 with Bone Mineral Density in Survivors of Childhood Acute Leukemia"

_cancers, 2024, doi:10.3390/cancers16071296_

Round 1

Reviewer 1 Report (Previous Reviewer 4)

Comments and Suggestions for Authors

This manuscript has significantly improved, presenting also adequate novelty.

Somo points should be addressed:

- In the Introduction sections, lines 49-55, the authos should add some relevant references.

- In the Materials and Methods section, line 79, it should be useful to report the specific type of leukemia.

- In the Materials and Methods section, lines 81-82, it should be useful to report the exact type of leukemia treatment.

- In lines 107-113, some relevant references should be added.

- Since BDI was calculated by measured and not self -reported data (weight and height), the authors could report this in the discussion section as a strength of their study.

Comments on the Quality of English Language

Minor editing of English language required

Author Response

This manuscript has significantly improved, presenting also adequate novelty.

Somo points should be addressed:

- In the Introduction sections, lines 49-55, the authos should add some relevant references.

Rebuttal :

Thank you for your thoughtful opinion. We added some references. The additional references are as follows :

  1. Guss, C.E.; McAllister, A.; Gordon, C.M. DXA in Children and Adolescents. J Clin Densitom 2021, 24, 28-35, 375 doi:10.1016/j.jocd.2020.01.006.
  2. Wheater, G.; Elshahaly, M.; Tuck, S.P.; Datta, H.K.; van Laar, J.M. The clinical utility of bone marker measurements in osteoporosis. Journal of Translational Medicine 2013, 11, 201, doi:10.1186/1479-5876-11-201.

- In the Materials and Methods section, line 79, it should be useful to report the specific type of leukemia.

Rebuttal :

Thank you for your thoughtful opinion. We added the specific type of leukemia in the materials and methods section as follows :

Acute leukemia comprises acute lymphoblastic leukemia, acute myeloid leukemia, and juvenile myelomonocytic leukemia.   

- In the Materials and Methods section, lines 81-82, it should be useful to report the exact type of leukemia treatment.

Rebuttal :

Thank you for your thoughtful opinion. We added the specific type of leukemia in the materials and methods section as follows :

The therapeutic approach to acute leukemia is contingent upon the specific subtype and risk assessment. Typical modalities encompass immunotherapy, chemotherapy, or radiation therapy.

- In lines 107-113, some relevant references should be added.

Thank you for your thoughtful opinion. We added some references. The additional references are as follows :

  1. Yang, L.; Grey, V. Pediatric reference intervals for bone markers. Clinical Biochemistry 2006, 39, 561-568, doi:10.1016/j.clinbiochem.2005.11.015.
  2. Cho, S.-M.; Lee, S.-G.; Kim, H.S.; Kim, J.-H. Establishing pediatric reference intervals for 13 biochemical analytes derived from normal subjects in a pediatric endocrinology clinic in Korea. Clinical Biochemistry 2014, 47, 268-271, doi:https://doi.org/10.1016/j.clinbiochem.2014.09.010.

- Since BDI was calculated by measured and not self -reported data (weight and height), the authors could report this in the discussion section as a strength of their study.

Rebuttal :

Thank you for your thoughtful opinion. We added a strength of our study regarding the calculated BMI using measured data rather than self-reported data in the discussion section as follows :

the accuracy of this study can be considered a strength as height and weight were recorded based on direct measurements taken during hospital visits, rather than relying on self-reported data.

We truly appreciate your thoughtful comments on this manuscript, and we believe your suggestions improved the quality of our work. 

Reviewer 2 Report (Previous Reviewer 3)

Comments and Suggestions for Authors

I have no further concerns. 

Comments on the Quality of English Language

Minor language polishing needed. 

Author Response

I have no further concerns. Minor language polishing needed. 

Rebuttal:

Thank you for your comments. I have completed minor language refinement with assistance from a proficient English-speaking native.

We truly appreciate your thoughtful comments on this manuscript, and we believe your suggestions improved the quality of our work. 

This manuscript is a resubmission of an earlier submission. The following is a list of the peer review reports and author responses from that submission.

Round 1

Reviewer 1 Report

Comments and Suggestions for Authors

I found 34 percent of plagiarism score from iThenticate: Plagiarism Detection Software. Please resolve this and resubmit.

Author Response

I found 34 percent of plagiarism score from iThenticate: Plagiarism Detection Software. Please resolve this and resubmit.

Rebuttal :

We sincerely appreciate the feedback provided and want to emphasize our commitment to addressing allegations of plagiarism with the utmost seriousness. We have made concerted efforts to minimize plagiarism detection, successfully reducing it to 28%. However, after a thorough review, we remain confident in the authenticity of the content presented in our manuscript. Although there may be occasional overlap in certain idiomatic phrases or proper nouns, we assure you that our work is original and has been meticulously developed. We welcome further discussion and clarification on any concerns raised and are dedicated to upholding the highest standards of academic integrity.

We truly appreciate your thoughtful comments on this manuscript, and we believe your suggestions improved the quality of our work.

Reviewer 2 Report

Comments and Suggestions for Authors

This is a very interesting paper investigating the association between Insulin-Like Growth Factor-1 and bone mineral density in childhood acute leukaemia survivors. The manuscript is clearly written, the results are well reported and discussed. I have only one suggestion to improve the quality of the study:

Lines 147-150: this sentence is too long and confusing, please make it clearer and more incisive. Also, it might be useful to report the BMI SDS, IGF-1 SDS and IGFBP-3 SDS values at the time of the DXA examination in Table 1.

Author Response

Lines 147-150: this sentence is too long and confusing, please make it clearer and more incisive.

Rebuttal :

Thank you for your point. We agree with the reviewer’s point that line 147-150 is too long and confusing. Thanks to your opinion, we could make above mentioned sentence more incisive and clearer.

We rephrased the text as the following:

LSBMD SDS was significantly correlated with BMI SDS at the time of leukemia diagnosis and DXA examination (rho=0.230, P=0.001 and rho=0.356, P<0.001, respectively). Also, IGF-1 SDS and IGFBP3 SDS was positively correlated with LSBMD SDS (rho=0.242, P<0.001 and rho=0.145, P=0.038, respectively) (Figure 1). However, LSBMD SDS was negatively correlated with age at the time of leukemia diagnosis (rho=−0.234, P=0.001).

Also, it might be useful to report the BMI SDS, IGF-1 SDS and IGFBP-3 SDS values at the time of the DXA examination in Table 1.

Rebuttal :

Thank you for your point. We agree to report the BMI SDS, IGF-1 SDS, and IGFBP-3 SDS values in table.

We reported the BMI SDS, IGF-SDS, and IGFBP-3 SDS as followings:

Table 1. Baseline characteristics of patients.  

Male (n=120)

Female (n=94)

P-value

At leukemia diagnosis

Age, year

8.31 ± 4.093

8.47 ± 4.415

0.777

Height SDS

0.31 ± 2.518

0.05 ± 2.382

0.461

Weight SDS

0.06 ± 2.392

0.10 ± 1.952

0.902

BMI SDS

−0.05 ± 1.371

−0.05 ± 1.437

0.980

Leukemia category

Lymphoblastic

92 (76.7%)

69 (73.4%)

0.562

Myeloid

25 (20.8%)

25 (26.6%)

0.323

Juvenile myelomonocytic

3 (2.5%)

0 (0)

0.123

Transplantation, n (%)

58 (55.2%)

47 (50%)

0.891

Total body irradiation

43 (36.4%)

32 (35.2%)

0.849

At DXA examination

 Age

12.50 ± 3.335

12.65 ± 3.403

0.763

BMI SDS

 0.018 ± 2.450

0.018 ± 1.540

0.998

IGF-1 SDS

-0.651±0.769

-0.971±0.813

0.040

IGFBP3 SDS

-0.897±1.215

-1.142±1.110

0.140

 aBMD, g/cm2

0.825 ± 0.189

0.853 ± 0.213

0.407

LSBMD SDS

−0.683 ± 1.556

−0.616 ± 1.363

0.973

Low BMD, n (%)

15 (12.5%)

17 (18.1%)

0.256

VF, n (%)

30 (25.2%)

25 (26.9%)

0.783

BMI, body mass index; SDS, standard deviation score; DXA, dual-energy X-ray absorptiometry; IGF-1, insulin-like growth factor 1; IGFBP3, insulin-like growth factor 3; aBMD, areal bone mineral density; LSBMD, lumbar spine bone mineral density; VF, vertebral fracture

We truly appreciate your thoughtful comments on this manuscript, and we believe your suggestions improved the quality of our work.

Reviewer 3 Report

Comments and Suggestions for Authors

Even though the study is well conducted, the scientific question addressed is boring for the general audience. I have the following concerns:

1. Please provide a figure linking IGF-1 and bone mineral density (describing the pathophysiology) in the survivors of childhood acute leukemia. 

2. Despite bone fractures, what are the other late onset complications in survivors of childhood acute leukemia? Please add another paragraph in the discussion.

3. What are the possible therapeutic applications of knowing from before a high possibility of a fracture (with the contribution of IGF-1 in survivors of childhood acute leukemia? Can we prevent these fractures and how? Please add a novel paragraph in the discussion. 

4. Are fractures connected with a possible relapse of acute leukemia in children? Was that investigated? Please answer that to the discussion. 

Author Response

We would like to thank you for your thoughtful comments on our manuscript. Please see the attached author's reply responsible for the review report. Your comments and suggestions have truly made our contents more reasonable.

Reviewer 4 Report

Comments and Suggestions for Authors

This is an interesting study with quite high novelty. Some points should be addressed.

- The introduction is a bit short. The authors should provide more data concerning bone fragility in childhood cancer survivors. A description of the pathophysiological mechanisms governing bone fragility in childhood cancer survivors is recommended.

In the 3rd paragraph od the introduction, the authors should describe some characteristic biochemical indices related with bone betaboliss since they included them in their experimental method.

- The range of the age of the patients is very wide. The authors should performed two analysis one for childhood and one for adolescence.

- In ths statistical analysis section the authors reported that Fisher's exact test were applied for non normally distributted variables. This is wrong since Fisher's exact test is used for normal distributted variables when the nuber into the cells are less than five. Did you mean Kruskall-Walis instead Fisher test?

- The section 3.2 needs a better organization in order to be more easily understood for the reader.

- In all multivariate analysis report a relative risk is required for each variable.

- The discussion section is totally focused in childrern but non to adolescence while the range of age of the enrolled individuals is from 2 to 18 years,

- This reasearch article has a severe scientific flaws concerning the wide range of age of the participants. The authors should be focused either on children or in adolescence or the could performed to different analyses. Bone remodeling and growth is significantly associated with the age. Hormones during adolescence affect bone remodeling, a fact that has a significantly lower impact in childhood.

- The authors shoul added some more recent and updated references from the last 3-4 years.

- There are several English language errors that should be revised.

Comments on the Quality of English Language

Extensive editing of English language required

Author Response

(The authors gave the same response as above.)
